# PFML: Self-Supervised Learning of Time-Series Data Without Representation Collapse

## Abstract

Self-supervised learning (SSL) is a data-driven learning approach that utilizes the innate structure of the data to guide the learning process. In contrast to supervised learning, which depends on external labels, SSL utilizes the inherent characteristics of the data to produce its own supervisory signal. However, one frequent issue with SSL methods is representation collapse, where the model outputs a constant input-invariant feature representation. This issue hinders the potential application of SSL methods to new data modalities, as trying to avoid representation collapse wastes researchers' time and effort. This paper introduces a novel SSL algorithm for time-series data called Prediction of Functionals from Masked Latents (PFML). Instead of predicting masked input signals or their latent representations directly, PFML operates by predicting statistical functionals of the input signal corresponding to masked embeddings, given a sequence of unmasked embeddings. The algorithm is designed to avoid representation collapse, rendering it straightforwardly applicable to different time-series data domains, such as novel sensor modalities in clinical data. We demonstrate the effectiveness of PFML through complex, real-life classification tasks across three different data modalities: infant posture and movement classification from multi-sensor inertial measurement unit data, emotion recognition from speech data, and sleep stage classification from EEG data. The results show that PFML is superior to a conceptually similar pre-existing SSL method and competitive against the current state-of-the-art SSL method, while also being conceptually simpler and without suffering from representation collapse.

## 1 Introduction

Self-supervised learning (SSL) can be described as a data-driven learning paradigm where the training process is guided by the inherent structure of the data itself. Unlike supervised learning that relies on externally provided labels, SSL exploits the intrinsic properties of the data to generate its own supervisory signal (Balestriero et al., 2023). SSL enables the model to learn rich feature representations from large amounts of unlabeled data that can be used as a starting point for downstream tasks, either as such or by fine-tuning the feature extractor to be better suited for solving some specific task (Erhan et al., 2010). Since typically there is an abundance of unlabeled data but a scarcity of labeled data, the use of SSL has been shown to reduce the need for large, manually annotated datasets (van den Oord et al., 2018; Baevski et al., 2020; Chen et al., 2020). In addition to SSL algorithms that have been developed for a single data modality, SSL algorithms that can be applied to multiple different data modalities have gained popularity in recent years (van den Oord et al., 2018; Akbari et al., 2021; Baevski et al., 2022; Wang et al., 2023). These methods and their extensions have shown great success in e.g. audio, image, and text data (van den Oord et al., 2018; Hénaff et al., 2020; Akbari et al., 2021; Baevski et al., 2022; Wang et al., 2023; Baevski et al., 2023; Yoon et al., 2023; Zhu et al., 2023; Lian et al., 2023).

However, many SSL algorithms suffer from two issues: First, SSL algorithms are usually complex, with a plethora of hyperparameters that need careful tuning for the algorithm to work properly. This hinders the ability of SSL algorithms to be applied to new data domains, where the selection of these hyperparameters is not self-evident. For example, in contrastive learning-based SSL, the selection of positive and negative samples during training is essential for the algorithm to work properly. However, deciding which samples should be assigned to positive and negative categories is not always apparent (Kalantidis et al., 2020; Robinson et al., 2021; Balestriero et al., 2023). As another example,

determining the number of clusters for clustering-based SSL algorithms (such as Caron et al. (2020) and Hsu et al. (2021)) in a new data domain or task can be difficult. Examples of such domains could include, for instance, different types of medical time-series data (e.g. EEG, ECG, or EMG recordings) that come in various dataset sizes and from various recording configurations. Second, a common failure mode during SSL pre-training is representation collapse, where the model ends up outputting a constant, time-invariant feature representation. Representation collapse is very common in SSL pre-training (Hua et al., 2021; Jing et al., 2022; Balestriero et al., 2023; Garrido et al., 2023), and many SSL methods apply different countermeasures to tackle the problem (see Section 3.1).

In the present study, we propose a new SSL algorithm for time-series data called Prediction of Functionals from Masked Latents (PFML). In PFML, the aim is to predict statistical functionals of the input signal corresponding to masked embeddings, given a sequence of unmasked embeddings. The overall methodological aim of our method is to have an SSL algorithm that would be as straightforward as possible to apply to various time-series data domains with minimal hyperparameter optimization, and without the risk of representation collapse. The contributions of the present study are as follows:

1. We propose a novel SSL algorithm for time-series data, PFML, that does not suffer from representation collapse, rendering the method straightforward to apply to new time-series data domains. To the best of our knowledge, PFML is the first work within the field of SSL for time-series data where the central idea of reconstructing statistical functionals is utilized.

2. We demonstrate the effectiveness of PFML using three different data modalities with complex, real-life classification tasks: infant posture and movement classification from multi-sensor inertial measurement unit (IMU) data, emotion recognition from speech data, and sleep stage classification from EEG data.

3. We show that PFML obtains both superior results against a conceptually similar pre-existing SSL method, and competitive results against the current state-of-the-art data modality agnostic SSL method, while also being conceptually simpler and without suffering from representation collapse.

## 2 RELATED WORK

Most of the advances in SSL have focused on developing new, better-performing algorithms with some specific data modality in mind. For speech data, Baevski et al. (2020) presented an SSL algorithm where the basic idea is to mask speech embeddings and then solve a contrastive task that is defined over a quantization of the embeddings which are simultaneously learned during the pre-training task. Hsu et al. (2021) proposed that instead of solving a contrastive task, they predict cluster targets of masked embeddings. Furthermore, the SSL method by Chen et al. (2022) also uses masking of embeddings, but the authors simulate noisy speech inputs and predict pseudo-labels of the original speech from the masked embeddings.

Similar to the advances in SSL for audio data, there have been significant developments in SSL for image data as well (Lee et al., 2017; Gidaris et al., 2018; Caron et al., 2018; Grill et al., 2020; Chen et al., 2020; Radford et al., 2021; He et al., 2022; Bao et al., 2022; Oquab et al., 2024). Grill et al. (2020) presented an SSL method that uses two neural networks that learn from each other's representations of differently augmented views of the same image. He et al. (2022) proposed masked autoencoders (MAE) that try to reconstruct masked patches of input images using an asymmetric encoder-decoder architecture. The SSL algorithm by Bao et al. (2022) tokenizes images into visual tokens, followed by masking some image patches and then trying to recover the original tokens from the masked patches.

SSL has also excelled in natural language processing (Devlin et al., 2019; Brown et al., 2020; Tay et al., 2023; OpenAI, 2023). Devlin et al. (2019) introduced an SSL method which obtains bidirectional feature representations from unlabeled text by conditioning on both the left and right textual context. The method by Brown et al. (2020) uses an autoregressive model which alternates dense and locally banded sparse attention patterns in their Transformer model. OpenAI (2023) proposed an expanded version of Brown et al. (2020) by making the model not only larger, but also capable of handling image inputs in addition to text inputs.

More recently, SSL literature has seen a growing number of work towards SSL algorithms capable of running the pre-training task on multiple different data modalities. The authors of van den Oord et al.

(2018) developed an SSL approach that predicts future embeddings based on previous context using contrastive learning. They showed that their method was able to learn useful feature representations for audio, image, text, and reinforcement learning in 3D environments. The SSL method by Akbari et al. (2021) also uses contrastive learning, but their method simultaneously takes audio, video, and text data as input and creates multimodal feature representations. These features were shown to work well with multiple different downstream tasks, i.e. video action recognition, audio event classification, image classification, and text-to-video retrieval. Wang et al. (2023) proposed an SSL method that performs prediction of masked tokens in a unified manner on images, texts, and image-text pairs. Their experiments showed that their method achieves state-of-the-art performance on various vision and vision-language tasks. Baevski et al. (2022) proposed data2vec, an SSL method for audio, image, and text data. In their approach, the model tries to predict masked latent features of an older version of itself that are both normalized and averaged over multiple Transformer layers. Their results in downstream tasks demonstrate the effectiveness of the method in all three data modalities.

For modality agnostic SSL algorithms, objective functions play a crucial role in guiding the learning process. These functions can be broadly categorized into three types: instance discrimination, clustering, and masked prediction. Instance discrimination aims to distinguish between different instances of data, thereby encouraging the model to learn unique features for each instance and enhancing the discriminative power of the learned representations. Contrastive learning methods, such as van den Oord et al. (2018); Baevski et al. (2020); He et al. (2020); Akbari et al. (2021); Pizzi et al. (2022), are an example of instance discrimination-based SSL methods. Clustering, on the other hand, groups similar instances together in the feature space, fostering the model to learn common features among instances belonging to the same group. Methods like Caron et al. (2020); YM. et al. (2020); Hsu et al. (2021) are examples of clustering-based SSL methods. Lastly, masked prediction involves the task of predicting masked parts of the input data based on the unmasked parts, thereby encouraging the model to learn contextual relationships within the data. Examples of such SSL methods include Devlin et al. (2019); Wang et al. (2020); Baevski et al. (2022); He et al. (2022); Xie et al. (2022).

## 3 METHOD

### 3.1 MOTIVATION

One key issue with many SSL methods is the problem of *representation collapse*, where the model outputs a constant, input-invariant feature representation, leading to a trivial solution of the pre-training task (Jing et al., 2022; Balestriero et al., 2023). This considerably slows down the development process for novel data domains and/or tasks due to the necessity of operating in uncertainty, when it is not clear whether the representation collapse is caused by an ill-posed task or by the SSL algorithm. To avoid this, SSL methods have taken several different countermeasures: Baevski et al. (2020) use the same target representations in their contrastive learning task in a dual manner, i.e. both as a positive and a negative example. Grill et al. (2020) both add an additional predictor to their training regime and use a moving average of their so-called online neural network to avoid representation collapse. Bardes et al. (2022) add a regularization term to their loss function that both maintains variance of the embeddings and decorrelates each pair of variables. In data2vec (Baevski et al., 2022), the authors tackle representation collapse by carefully selecting their model hyperparameters and promoting target representation variance through feature normalization. Also, in the code implementation of data2vec[1], pre-training is stopped if the variance of either model predictions or training targets falls below a predefined threshold.

Intuitively, given a trivial task, the model does not learn useful feature representations during pre-training. In contrast, if the learning objective is too complicated, the model fails to converge to a useful solution. For time-series data, i.e. a waveform (e.g. audio) or a set of waveforms (e.g. multi-channel EEG), trying to reconstruct masked parts of the input signal given the unmasked parts of the signal (as in e.g. MAE (He et al., 2022)) is a very complex task. This is due to the fact that a time-series signal can have large temporal variation even between short periods of time. While joint learning of *a priori* unspecified latent representations and their prediction allows discarding of this irrelevant variation (as in, e.g., van den Oord et al. (2018) or Baevski et al. (2020)), the problem

---

[1]`https://github.com/facebookresearch/fairseq/tree/main/examples/data2vec`

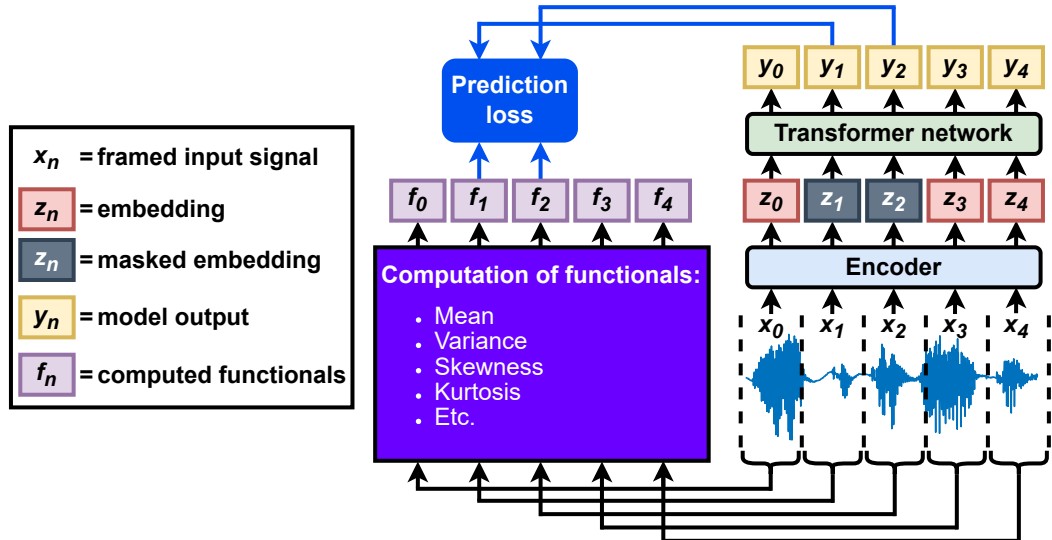

Figure 1: An overview of the PFML pre-training pipeline.

requires learning algorithms that become susceptible to representation collapse and/or may require careful tuning of the training process. We hypothesize that for SSL pre-training with time-series data, a model would learn more useful features for downstream tasks if the complex setting of MAE would be alleviated slightly. Hence, we propose Prediction of Functionals from Masked Latents (PFML), a novel SSL algorithm for time-series data. Our method builds on the concept of MAE and reduces the complexity of the pre-training task of MAE in two ways:

1. Instead of aiming to reconstruct the input signal, the model tries to predict a set of statistical functionals computed from the input signal.

2. Instead of masking the input signal directly, PFML borrows the idea of e.g. wav2vec 2.0 (Baevski et al., 2020) and data2vec (Baevski et al., 2022) and masks the embeddings created by the encoder model.

Regarding point (1), by making the model predict statistical functionals of masked latent features instead of predicting the input signal $\mathbf{x}$ itself, we relieve the model from the complex task of modelling the high-dimensional distribution of $\mathbf{x}$ in detail. We validate this argument of generating better features for downstream tasks by reducing the computational complexity of the pre-training task in Section 4, where we compare our proposed method against MAE. In theory, the set of statistical functionals can be chosen so that the desired and deterministically calculated statistical properties of the data, and thereby their variance, are preserved in the target features. Furthermore, regarding point (2), we show in our experiments in Section 4 that it is more beneficial during pre-training to mask the latent features instead of masking the input directly. This further alleviates the complexity of the learning task in particular for the encoder module.

### 3.2 PREDICTION OF FUNCTIONALS FROM MASKED LATENTS

Figure 1 depicts an overview of the PFML pre-training pipeline. First, a single- or multi-channel signal $\mathbf{x}$ is framed into a sequence of short-term frames $\{\mathbf{x}_0, \mathbf{x}_1, ...\}$, $\mathbf{x}_n = \{x_t, x_{t+1}, ..., x_{t+N-1}\}$, of $N$ samples each. Then, a set of $m$ *functionals*, $\mathcal{F} = \{F_0, F_1, ..., F_{m-1}\}$, is computed for each frame $\mathbf{x}_n$ to produce corresponding functional values $\mathbf{f}_n = \{F_0(\mathbf{x}_n), F_1(\mathbf{x}_n), ..., F_{m-1}(\mathbf{x}_n)\}$. Here, functionals are defined as mathematical operations which map a time series of arbitrary length into a single value, such as the mean or variance of the signal. The frames $\mathbf{x}_n$ are also fed to an encoder model, which converts the framed signals into embeddings $\mathbf{z}_n$. Some of these embeddings are masked randomly at time steps $M$ (for example, $M \in \{1, 2\}$ in Figure 1), after which all $\mathbf{z}_n$ are used as an input for a Transformer-based model to obtain outputs $\mathbf{y}_n$. Finally, a prediction loss is computed between the outputs of masked time steps $\mathbf{y}_M$ and their functional counterparts $\mathbf{f}_M$. As a result,

PFML pre-training optimizes the prediction of functionals of input signal frames corresponding to the masked embeddings, given the unmasked embeddings from the temporal context of these frames.

In PFML, predicting only one or a few functionals of a framed signal can be a trivial task, and will most probably lead to learning feature representations that are not very useful for downstream tasks. However, as the number of functionals that each describe some property of the framed signal grows, a more accurate description of the signal can be obtained (see e.g. McDermott & Simoncelli (2011) for representing perceptual properties of sound textures with functionals). Therefore, as the number of different functionals grows, the PFML algorithm is getting closer to predicting all of the nuances of the input signal.

Let us assume the following in PFML pre-training:

- Assumption 1: There is temporal variability across the frames $\mathbf{x}_n$. This assumption is reasonable as real-world data typically exhibits temporal variability.
- Assumption 2: Given Assumption 1, a set of non-trivial functionals $\mathcal{F}$ computed from $\mathbf{x}_n$ also contains variance across the frames. This follows naturally since non-constant functionals derived from variable data also exhibit variability.

Under these assumptions, as the model is trying to predict the computed functionals $\mathbf{f}_n$ given the embeddings $\mathbf{z}_n$, good model predictions $\mathbf{y}_n$ that lead to low prediction loss values also inherently contain variance. On the contrary, if $\mathbf{y}_n$ were to contain zero variance across the frames while $\mathbf{f}_n$ contains variance, the prediction loss would be high. Consequently, PFML pre-training does not converge to collapsed feature representations, as long as Assumptions 1 and 2 hold true. For a more detailed formulation, see Appendix A. Empirical results (see Section 4.4) support this theoretical claim, showing that PFML maintains variance in predictions across various datasets.

In the present study, we selected 11 mathematical operations as our set of functionals: mean, variance, skewness, kurtosis, minimum value, maximum value, zero-crossing rate (ZCR), and the mean, variance, skewness, and kurtosis of the autocorrelation function (ACF). The ZCR for a signal $\mathbf{x} = \{x_0, x_1, ..., x_{N-1}\}$ is defined as

$$\mathrm{ZCR}(\mathbf{x}) = \frac{1}{N-1} \sum_{k=1}^{N-1} |\mathrm{sgn}(x_k) - \mathrm{sgn}(x_{k-1})| , \tag{1}$$

where *sgn* denotes the sign function (Rabiner & Schafer, 2007). The ACF for a signal $\mathbf{x}$ at lag $\tau$ is defined as

$$\mathrm{ACF}(\mathbf{x}, \tau) = \frac{1}{(N-\tau)\sigma^2} \sum_{k=0}^{N-\tau-1} (x_{k+\tau} - \mu)(x_k - \mu) , \tag{2}$$

where $\tau < N$, $\mu$ is the mean of $\mathbf{x}$, and $\sigma^2$ is the variance of $\mathbf{x}$ (Rabiner & Schafer, 2007). Note that Equation 2 returns a vector of measurements when applied to all lags $\tau < N$.

For masking the embeddings, in each training and validation minibatch we randomly select frames with a probability of $p_m$ to be mask starting indices, and we mask the embedding of that frame and $m_l - 1$ subsequent frames, resulting in a minimum mask length of $m_l$ frames. We replace each embedding that is selected for masking with a vector of ones. Masks can overlap, enabling longer mask spans than $m_l$ frames (especially with high $p_m$). Furthermore, we also define that each training and validation sequence needs to have at least one mask starting index during PFML pre-training.

Note that the PFML pre-training process is not restricted to any specific type of neural networks. In the present study, we used convolutional neural networks (CNNs) as our encoder model, and $T$ Transformer encoder blocks as the temporal model. However, any type of encoder could be used for PFML, as long as the encoder can convert time-series data into a sequence of embeddings. Furthermore, other temporal models, such as conformer-based models (Gulati et al., 2020) or bidirectional recurrent neural networks (Hochreiter & Schmidhuber, 1997; Cho et al., 2014), could also be used for PFML, as long as the model is able to take contextual information into account.

## 4 EXPERIMENTS

We evaluate our PFML method using three different datasets of time-series data with complex classification tasks: infant posture and movement classification from multi-sensor IMU data, emotion

recognition from speech data, and sleep stage classification from EEG data. For each dataset, we first run SSL pre-training with unlabeled data using PFML, after which we fine-tune our models for downstream classification tasks using labeled data. We compare PFML against three different baselines: MAE (He et al., 2022), data2vec (Baevski et al., 2022), and not using pre-training at all. We selected MAE for our experiments since it is conceptually very similar to PFML, and we chose data2vec since it is the current state-of-the-art data modality agnostic SSL method. In order to make the prediction of functionals directly comparable with predicting the input signal, we use a slightly modified version of MAE where we mask embeddings instead of masking inputs.

For PFML pre-training, our models consist of a modality-specific frame-level encoder (detailed in Sections 4.1, 4.2, and 4.3 for IMU, speech, and EEG data, respectively) and a Transformer network consisting of $T$ Transformer encoder blocks. Between the encoder and Transformer networks there is a CNN-based relative positional encoder followed by a GeLU (Hendrycks & Gimpel, 2016) activation and layer normalization (Ba et al., 2016). We frame our input signals before feeding the data into an encoder model, and we compute functionals from these frames as our training targets. For multi-channel data, we compute functionals separately for each channel. The functionals are then z-score normalized across the entire pre-training dataset. For computational efficiency, we pre-compute the functionals of each signal frame before the pre-training process. After the Transformer encoder blocks, we add a linear projection to convert the Transformer outputs into predicted functionals. Pre-training is run until validation loss convergence, and we use the model with the lowest validation loss as our pre-trained model. Starting from an initial learning rate, we gradually reduce the learning rate during model training with a reduction factor of 0.5 based on the plateauing of the validation loss.

We pre-train our models using MAE and data2vec in a similar manner as for PFML, and we use the same model architecture for all three pre-training algorithms. MAE pre-training is run in a similar manner as PFML pre-training, with the only exception of predicting the input signal frames instead of functionals. For data2vec pre-training, we used the instance-normalized (Ulyanov et al., 2016) and averaged outputs of each feed-forward part of all Transformer encoder blocks as our training targets. If we observed that a representation collapse occurred during data2vec pre-training, we restarted the pre-training process. For further details on the data2vec algorithm, see Baevski et al. (2022). We used mean squared error loss for all pre-training processes except for PFML with speech data, where we found L1 loss to work better.

We fine-tune our pre-trained models in two stages. In the first stage, two randomly initialized fully-connected GeLU layers followed by a softmax function are added after the Transformer model. Then, these layers are fine-tuned separately as the weights of the encoder and Transformer are frozen. In the second stage, the entire model is fine-tuned with the same hyperparameters as in the first fine-tuning stage with one exception: The learning rate $LR$ is linearly increased from $0.001 \cdot LR$ to $LR$ during a warm-up period of 20 training epochs, followed by reduction by a factor of 0.5 based on validation loss plateauing. We use weighted categorical cross-entropy loss by weighting the loss of each sample by its class' inverse frequency.

We also test the linear separability of the features learned by our pre-trained models. In this case, we only add one linear layer after the Transformer model, and we fine-tune this single layer while the weights of the encoder and Transformer are frozen. As a baseline, we perform the same linear evaluation for a randomly initialized model without any pre-training.

For pre-training and fine-tuning, we use the RAdam (Liu et al., 2020) and Adam (Kingma & Ba, 2015) optimizers, respectively. For the "no pre-training" condition, we simply omit pre-training, the first fine-tuning stage, and the learning rate warm-up period of the second fine-tuning stage. We used an NVIDIA Tesla V100 GPU to train our models, and we implemented the code using PyTorch version 1.13.1. Our implementation is publicly available on GitHub.[2]

In order to demonstrate the superiority of PFML against the state-of-the-art SSL method for multiple data modalities, data2vec, in terms of representation collapse, we ran PFML, MAE, and data2vec pre-training 10 times using the best hyperparameter combinations for each SSL method and for each data modality. We defined representation collapse to have occurred if the variance of either the embeddings or model outputs fell below 0.01 for 10 consecutive pre-training epochs, during which the validation loss was decreasing. In our preliminary experiments, we found that this condition was

---

[2](Here will be a link to our GitHub repository.)

a good indicator of an upcoming representation collapse: A systematic decrease in the variance of a model's embeddings or outputs indicates impending representation collapse in SSL methods where the model can invent its own training targets.

## 4.1 Infant Posture and Movement Classification

For infant posture and movement classification, we use the multi-sensor IMU data from Airaksinen et al. (2022). The data contains 24-channel signals from infants (three gyroscope and three accelerometer channels, four limbs) with a sampling rate of 52 Hz. We window the signals into 120-sample frames (approx. 2.3 seconds) with 50% overlap. For further details about the dataset, see Airaksinen et al. (2022).

For model pre-training, we use a 387-hour set of unlabeled IMU data from infant free-form play that has been automatically screened for signal quality (Vaaras et al., 2023b). This subset contains 4669 sequences of 260 consecutive frames, each corresponding to five minutes of data. As the encoder, we use the same four-layer CNN-based encoder architecture as in Airaksinen et al. (2022) with three minor modifications that were found to improve training efficiency and system performance when replicating the experiments of Airaksinen et al. (2022): We added layer normalization after the last two convolutions to make the pre-training process more stable, the kernel size of the second convolutional layer of the CNN encoder was changed from [4,5] to [3,5], and the originally temporally asymmetrical padding was set to [1,2] to make it symmetric. The pre-training data is randomly split into a training and validation set in a ratio of 80:20 sequences, and we input 260-frame sequences into the model.

For fine-tuning our pre-trained models, we use a 29-hour (91,449 frames) labeled dataset of IMU data (41 recordings and distinct participants) for two separate tasks: posture classification and movement classification. The data contains nine annotated movement categories (still, roll left/right, pivot left/right, proto/elementary/fluent movement, transition) and seven annotated posture categories (prone, supine, left/right side, crawl posture, sitting, standing) for each 2.3-second frame. For model training, we use all annotated data, but we only use the frames in which all annotators agreed on the label for model testing. We train our models separately for both classification tasks using the so-called iterative annotation refinement labels from Airaksinen et al. (2020).

Model fine-tuning is run using recording-level 10-fold cross-validation on the 41 distinct recordings of the labeled dataset. We split each training fold into separate training and validation sets in a ratio of 80:20 recordings. The unweighted average F1 score (UAF1) on the validation set is used as the training criterion, and we select the best-performing model based on validation set UAF1 score. We use random sensor dropout ($p = 0.3$) for data augmentation during model fine-tuning. The final UAF1 score of fine-tuning is computed from an aggregate confusion matrix across all test folds. For further details regarding the pre-training and fine-tuning hyperparameters, see Appendix B.

## 4.2 Speech Emotion Recognition

We use the 56-hour subset of Finnish speech of the NICU-A corpus (Vaaras et al., 2023a) for our speech emotion recognition experiments. This subset contains 129,007 utterances with a sampling rate of 16 kHz, of which 5198 and 345 belong to annotated training and testing sets, respectively. Each annotated utterance in NICU-A contains binary labels for emotional valence (positive/non-positive) and arousal (high/low). We window each speech signal into 30-ms frames with a 20-ms overlap. Each sequence is z-score normalized, and we zero-pad or truncate each normalized sequence into 3-second segments (301 frames). See Vaaras et al. (2023a) for further details on NICU-A.

For model pre-training, we use all 129,007 utterances, and we input 301-frame sequences to our model. We use a four-layer CNN encoder with output channels [128, 128, 128, 128], kernel sizes [10, 8, 4, 4], strides of [5, 4, 2, 2], and paddings of [3, 2, 1, 1]. Each layer is followed by layer normalization, a GeLU nonlinearity, and dropout. The last CNN layer is followed by average pooling with a kernel size of 6 before dropout. The pre-training utterances are randomly split into a training and validation set in a ratio of 80:20 sequences.

We fine-tune and test our models separately for both classification tasks (valence/arousal) using the labeled 5198- and 345-utterance training and testing sets, respectively. The training set is randomly split into a training and validation set in a ratio of 80:20 utterances, and we select the best-performing

Table 1: Downstream task fine-tuning results for PFML, data2vec, MAE, and not using pre-training at all for the five different classification tasks across the three different data modalities (IMU, speech, and EEG data).

| | Multi-sensor IMU data (infant motility assessment) | | Speech data (speech emotion recognition) | | EEG data (sleep stage classification) |
|---|---|---|---|---|---|
| | Movement | Posture | Valence | Arousal | Sleep stage |
| No pre-training | 80.6 | 94.9 | 68.2 | 65.5 | 69.1 |
| MAE | 81.0 | 95.6 | 69.9 | 68.1 | 70.5 |
| data2vec | **81.9** | **95.8** | **70.7** | 68.5 | 69.8 |
| PFML (ours) | 81.8 | 95.7 | **70.7** | **68.6** | **71.2** |
| | UAF1 (%) | | UAR (%) | | UAF1 (%) |

model of the fine-tuning process based on the unweighted average recall (UAR) performance score on the validation set. This model is then used to compute the UAR performance score of the test set. See Appendix B for further details regarding the pre-training and fine-tuning hyperparameters.

### 4.3 SLEEP STAGE CLASSIFICATION

For sleep stage classification, we use the pre-processed expanded Sleep-EDF Database (Kemp et al., 2000; Goldberger et al., 2000) from a study by Eldele et al. (2021). The dataset contains 30-second segments of the Fpz-Cz channel with a sampling rate of 100 Hz, comprising a total of 195,479 segments of EEG data. Each 30-second segment belongs to one of five annotated categories: wake, rapid eye movement (REM), non-REM stage 1, non-REM stage 2, or non-REM stages 3 and 4 combined. We z-score normalize each 30-second segment, and we window each segment into 4-second frames with 2 seconds of overlap, resulting into 14 frames for each segment.

We pre-train our models using all 195,479 EEG segments. We use the 14-frame sequences as our input for a three-layer CNN encoder with output channels $[128, 128, 128, 128]$, kernel sizes $[10, 8, 4]$, strides of $[5, 5, 3]$, and paddings of $[3, 2, 1]$. Each convolution is followed by layer normalization, a GeLU nonlinearity, and dropout. The third CNN layer is followed by average pooling with a kernel size of 5 before dropout. We randomly split the EEG segments for pre-training into a training and validation set in a ratio of 80:20 segments.

We fine-tune our models for sleep stage classification using 10-fold cross-validation at the test subject-level on the 78 test subjects of the dataset. Each training fold is split into training and validation sets at the test subject-level in a ratio of 80:20 test subjects. Similar to Sec. 4.1, we use the validation UAF1 score as our training criterion, and the testing UAF1 score is computed from an aggregate confusion matrix across all test folds. For further details on the training hyperparameters, see Appendix B.

### 4.4 RESULTS

Table 1 presents the fine-tuning results of the comparison of our PFML method against MAE, data2vec, and not using pre-training at all. Across all three data modalities and five classification tasks, the results show that PFML outperformed MAE and achieved highly comparable results to data2vec. Using pre-training with any SSL method provided superior results as opposed to not using pre-training at all. For the classification of posture from IMU data, there were only minor differences in performance between different SSL methods. In sleep stage classification from EEG data, both MAE and PFML outperformed data2vec by a large margin. The comparison between PFML and MAE showcases that it is more beneficial to predict functionals than to predict the input signal.

Table 2 shows the results of the linear evaluation experiments. Similar to the results of Table 1, PFML outperformed MAE and was comparable to data2vec when using the pre-trained models as feature extractors for linear classifiers. Again, both MAE and PFML outperformed data2vec by a large margin in sleep stage classification from EEG data. In the case of using a randomly initialized model as a feature extractor for linear classifiers, the classification accuracy was at chance-level in all cases except when classifying posture for IMU data.

The results of representation collapse experiments are shown in Table 3. As can be seen from the results, it is very common for representation collapse to occur with data2vec across all data modalities.

Table 2: Linear evaluation results for PFML, data2vec, MAE, and a randomly initialized model.

| | Multi-sensor IMU data (infant motility assessment) | | Speech data (speech emotion recognition) | | EEG data (sleep stage classification) |
|---|---|---|---|---|---|
| | Movement | Posture | Valence | Arousal | Sleep stage |
| *Random initialization* | 10.8 | 45.9 | 51.4 | 50.8 | 20.6 |
| *MAE* | 39.9 | 87.2 | 60.9 | 58.8 | 43.7 |
| *data2vec* | 41.7 | 87.1 | **61.8** | **59.3** | 41.5 |
| *PFML (ours)* | **43.8** | **87.4** | 61.6 | 59.2 | **44.1** |
| | UAF1 (%) | | UAR (%) | | UAF1 (%) |

Table 3: Frequency of representation collapse across 10 runs of PFML, data2vec, and MAE for each tested data modality.

| | Multi-sensor IMU data | Speech data | EEG data |
|---|---|---|---|
| *MAE* | **0/10** | **0/10** | 1/10 |
| *data2vec* | 9/10 | 8/10 | 8/10 |
| *PFML (ours)* | **0/10** | **0/10** | **0/10** |

On the contrary, the results indicate that MAE and PFML do not suffer from representation collapse: PFML did not experience representation collapses at all, and MAE had a representation collapse only once. Furthermore, we attribute this single representation collapse of MAE to bad luck in model weight initialization, as in this particular case the model loss started diverging from the beginning of the pre-training process. The results showcase that methods like MAE and PFML, whose training targets inherently contain variance, are less prone to representation collapse compared to methods like data2vec that learn their own prediction targets.

## 4.5 ADDITIONAL HYPERPARAMETER EXPERIMENTS

In order to demonstrate that it is more beneficial during pre-training to mask the latent features instead of masking the input directly, we ran PFML pre-training for all three datasets twice: either by masking the inputs or by masking the embeddings. Subsequently, we fine-tuned our models for all five classification tasks, and the results are shown in Table 6 of Appendix C. As can be observed from the results, it is more beneficial for downstream tasks if we alleviate the complexity of the pre-training task for the encoder by masking the embeddings instead of masking the inputs. The only exception was with EEG data, where it did not make a difference whether inputs or embeddings were masked.

For each data modality, we also experimented with different configurations of masking probability $p_m$ and the length of the masks $m_l$. We ran PFML pre-training using different configurations of $p_m$ and $m_l$, and then we fine-tuned the pre-trained models. For IMU and speech data, we only experimented with one classification task each, namely classification of movement from IMU data and classification of valence from speech data. The results for different configurations of $p_m$ and $m_l$ for IMU, speech, and EEG data are shown in Appendix C in Tables 7, 8, and 9, respectively. For IMU data, the differences between different masking strategies are rather small, whereas for speech and EEG data the selection of masking hyperparameters has a notable effect on fine-tuning performance.

We also experimented with the effect of discarding some of the functionals in PFML pre-training for IMU data. After pre-training, we fine-tuned our model for movement classification, and the results are presented in Table 10 of Appendix C. The results indicate that using the full set of 11 functionals during PFML pre-training provides the best outcome. As the number of discarded functionals increases, the prediction task becomes simpler and the training targets are able to capture less information of the input signal frames, leading to worse fine-tuning performance.

Finally, we tested different mask types for PFML pre-training using IMU data. We either replaced the masked embeddings with a fixed vector of zeros, ones, random Gaussian noise (as in e.g. Baevski et al. (2023)), or a learnable mask token (as in e.g. Baevski et al. (2022)). After PFML pre-training using the four different mask types, we fine-tuned the pre-trained models for movement classification. Table 11 of Appendix C presents the comparison results for different mask types. As can be observed, the choice between a mask of ones or random Gaussian noise does not have a notable impact on the performance. However, using a learnable mask token yielded slightly worse results than a vector

of ones or random Gaussian noise, and a vector of zeros yielded the worst results. We observed that either using a vector of ones, random Gaussian noise, or learnable mask tokens for masking the embeddings promoted embedding variance, whereas using a vector of zeros provided a smaller level of variance for the embedding representations during pre-training. This lower level of variance for embeddings might potentially hinder the fine-tuning process, resulting into a lower performance in downstream tasks.

## 5 CONCLUSION

In this paper, we presented PFML, a novel SSL algorithm for time-series data that avoids the common SSL issue of representation collapse. PFML operates by predicting statistical functionals of the input signal corresponding to masked embeddings, given a sequence of unmasked embeddings. We demonstrated the effectiveness of PFML using five different classification tasks across three different data modalities: infant posture and movement classification from multi-sensor IMU data, emotion recognition from speech data, and sleep stage classification from EEG data. Our results show that PFML is superior to a conceptually similar SSL method, MAE. Our results also show that PFML is competitive against the current state-of-the-art data modality agnostic SSL method, data2vec, while being conceptually simpler and without suffering from representation collapse. The fact that PFML matches the performance of data2vec while also avoiding the issue of representation collapse renders PFML more straightforward to apply to new time-series data domains, such as in the case of clinical time-series data. The present work may also be extended to other domains than time-series data, such as images where functionals could be computed of, e.g., image patches.

**Limitations** We selected the present set of 11 functionals for their effectiveness across the three data modalities used in the present study, aiming for potential generalizability and a robust starting point to other data domains and downstream tasks. However, carefully selecting the number and type of functionals specifically for different modalities may lead to better results than presented here. Also, we did not include data augmentation in our pre-training processes to save computational time for PFML pre-training, as we wanted to pre-compute the functionals before the model training. As shown in e.g. Chen et al. (2020); Grill et al. (2020); He et al. (2022); Balestriero et al. (2023), data augmentation during pre-training may lead to improved performance on downstream tasks. Nonetheless, performing masking for randomly sampled frames is already a form of data augmentation in itself. Furthermore, other model architectures besides CNN-based encoders or Transformer encoder blocks could also be used, and this may improve PFML pre-training performance. Lastly, we acknowledge that typically SSL pre-training is run with very large minibatch sizes using multiple GPUs, and the results of the present experiments might improve with larger minibatch sizes. However, to promote reproducibility and encourage other researchers to try PFML, we deliberately pre-trained our models using relatively small minibatches so that the pre-training processes could be run on a single GPU with 16 GB of VRAM. As detailed in Appendix D, our method used only a moderate amount of computational resources.

**Broader Impacts** Since the main goal of PFML is to make the algorithm straightforwardly applicable to different time-series data domains, our method makes it easier to apply SSL pre-training for time-series data without complex tuning of hyperparameters or the need to profoundly understand the target data domain. As an example, properties of different medical time-series data, such as those obtained with EEG, ECG, or EMG, can be dependent on the clinical environment, the specific measurement equipment and setup, or clinical population being measured (Watson et al., 2019). This limits the applicability of 'universal' pre-trained models predominant in computer vision and speech technology. In a similar manner, various industrial sensor setups, such as those for system monitoring and predictive maintenance (accelerometers, magnetometers etc.), can result in data unique to a particular environment or machine type. In these cases, the use of PFML pre-training can be practical, since applying modality-specific SSL algorithms or fine-tuning pre-trained models from other data modalities might not generalize well to novel time-series data domains. Hence, PFML may promote the use of machine learning as an assisting tool in e.g. clinical healthcare or other limited-data domains. However, as with all classifiers, machine-learning models trained using PFML might make errors. Incorrect model-based decisions, such as incorrect diagnoses, may be detrimental in some cases. Lastly, any bias, private information, or harmful content in the pre-training data can, in theory, be reflected to the feature representations that are learned by PFML.

REPRODUCIBILITY STATEMENT

In order to promote reproducibility, we provide the implementation of the PFML algorithm for all three data modalities that were used in the present study (IMU, speech, and EEG data) in GitHub: (link here). Also, all experimental steps are described in detail in Section 4, and the hyperparameters used in both model pre-training and fine-tuning are listed in Appendix B. Furthermore, to encourage other researchers to try PFML, we pre-trained our models using relatively small minibatches so that the pre-training processes could be run using single-GPU setups. **We will provide a link to an anonymous repository containing our code implementation for the reviewers and ACs once the discussion forums are opened in OpenReview.**

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

## A    PROOF OF NON-COLLAPSED FEATURE REPRESENTATIONS IN PFML PRE-TRAINING

This section provides a more detailed mathematical formulation for the proof that PFML pre-training does not converge to collapsed feature representations.

Let $\mathbf{x}$ be a single- or multi-channel time-series signal, framed into a sequence of short-term frames $\{\mathbf{x}_0, \mathbf{x}_1, ...\}$ of $N$ samples each, where $\mathbf{x}_n = \{x_t, x_{t+1}, ..., x_{t+N-1}\}$. We define a set of $m$ functionals, $\mathcal{F} = \{F_0, F_1, ..., F_{m-1}\}$, to be computed for each frame $\mathbf{x}_n$ to produce a set of computed functionals $\mathbf{f}_n$. Here, we refer to functionals as mathematical operations which map a time series of arbitrary length into a single value, such as the mean or variance of the signal. Also, let $\mathbf{z}_n$ be the output embeddings of an encoder model given the input $\mathbf{x}_n$, and let $\mathbf{y}_n$ denote the output predictions of a Transformer-based model given the input $\mathbf{z}_n$.

To formalize the relationships between inputs and outputs, let us define the following functions:

- Let $\mathcal{F}$ be the set of functionals that maps the input frames $\mathbf{x}_n$ to the computed functionals $\mathbf{f}_n$, i.e., $\mathbf{f}_n = \mathcal{F}(\mathbf{x}_n) = \{F_0(\mathbf{x}_n), F_1(\mathbf{x}_n), ..., F_{m-1}(\mathbf{x}_n)\}$.
- Let $g$ be the function that maps the embeddings $\mathbf{z}_n$ to the predictions $\mathbf{y}_n$, i.e., $\mathbf{y}_n = g(\mathbf{z}_n)$.

Let us assume the following in PFML pre-training:

- Assumption 1: There is temporal variability across the frames $\mathbf{x}_n$. Formally, let $\sigma^2(\mathbf{x}_n)$ denote the variance of $\mathbf{x}_n$ across the frames, and $\sigma^2(\mathbf{x}_n) > 0$.
- Assumption 2: Given Assumption 1, the set of non-trivial functionals $\mathcal{F}$ computed from $\mathbf{x}_n$ also contains variance across the frames. Formally, let $\sigma^2(\mathbf{f}_n)$ denote the variance of $\mathbf{f}_n$ across the frames, and $\sigma^2(\mathbf{f}_n) > 0$.

Under these assumptions, we aim to show that the predictions $\mathbf{y}_n$ also contain variance across the frames, i.e., $\sigma^2(\mathbf{y}_n) > 0$.

In PFML pre-training, the model learns to predict the computed functionals $\mathbf{f}_n$ given the embeddings $\mathbf{z}_n$. The prediction loss $L$ is defined as

$$L = \frac{1}{N} \sum_{n=1}^{N} (\mathbf{f}_n - \mathbf{y}_n)^2 \tag{3}$$

for the MSE loss and

$$L = \frac{1}{N} \sum_{n=1}^{N} |\mathbf{f}_n - \mathbf{y}_n| \tag{4}$$

for the L1 loss.

To minimize either the MSE or L1 loss functions (Equations 3 and 4, respectively), the predictions $\mathbf{y}_n$ must closely match the computed functionals $\mathbf{f}_n$. If $\mathbf{y}_n$ were to contain zero variance across the frames, i.e., $\sigma^2(\mathbf{y}_n) = 0$, while $\mathbf{f}_n$ contains variance, i.e., $\sigma^2(\mathbf{f}_n) > 0$, the prediction loss $L$ would be high. This is because the constant predictions $\mathbf{y}_n$ would not be able to capture the temporal variability in $\mathbf{f}_n$.

Therefore, to achieve low prediction loss values, the predictions $\mathbf{y}_n$ must also contain variance across the frames, i.e., $\sigma^2(\mathbf{y}_n) > 0$. Consequently, PFML pre-training does not converge to collapsed feature representations, as long as Assumptions 1 and 2 hold true. $\square$

Given that real-world time-series data generally shows temporal variability, and computed functionals derived from such data are expected to reflect this variability, Assumptions 1 and 2 are valid for most real-world datasets.

## B    PRE-TRAINING AND FINE-TUNING HYPERPARAMETERS

This section provides details on the pre-training (Table 4) and fine-tuning (Table 5) hyperparameters of the present experiments.

Table 4: The pre-training hyperparameters for PFML, data2vec, and MAE pre-training for each data modality (IMU, speech, and EEG data).

| | Multi-sensor IMU data | | | Speech data | | | EEG data | | |
|---|---|---|---|---|---|---|---|---|---|
| | **PFML** | **data2vec** | **MAE** | **PFML** | **data2vec** | **MAE** | **PFML** | **data2vec** | **MAE** |
| Patience (epochs) | 100 | 100 | 100 | 25 | 25 | 25 | 25 | 25 | 25 |
| Initial LR | 1e-4 | 1e-4 | 1e-4 | 1e-4 | 1e-4 | 1e-4 | 1e-4 | 1e-4 | 1e-4 |
| LR scheduler patience (epochs) | 40 | 40 | 40 | 10 | 10 | 10 | 10 | 10 | 10 |
| LR scheduler reduction factor | 0.5 | 0.5 | 0.5 | 0.5 | 0.5 | 0.5 | 0.5 | 0.5 | 0.5 |
| Optimization algorithm | RAdam | RAdam | RAdam | RAdam | RAdam | RAdam | RAdam | RAdam | RAdam |
| Minibatch size | 64 | 64 | 64 | 64 | 64 | 64 | 1024 | 1024 | 1024 |
| Loss function | MSE | MSE | MSE | L1 | MSE | MSE | MSE | MSE | MSE |
| Dropout (encoder) | 0.1 | 0.1 | 0.1 | 0.2 | 0.2 | 0.2 | 0.1 | 0.1 | 0.1 |
| Dropout (Transformer) | 0.2 | 0.2 | 0.2 | 0.2 | 0.2 | 0.2 | 0.2 | 0.2 | 0.2 |
| Activation function (Transformer) | GeLU | GeLU | GeLU | GeLU | GeLU | GeLU | GeLU | GeLU | GeLU |
| Input/output dim (Transformer) | 160 | 160 | 160 | 128 | 128 | 128 | 128 | 128 | 128 |
| Num encoder blocks (Transformer) | 6 | 6 | 6 | 6 | 6 | 6 | 6 | 6 | 6 |
| Feed-forward inner dim (Transformer) | 640 | 640 | 640 | 512 | 512 | 512 | 512 | 512 | 512 |
| Num attention heads (Transformer) | 10 | 10 | 10 | 8 | 8 | 8 | 8 | 8 | 8 |
| Relative positional encoding kernel size (Transformer) | 13 | 13 | 13 | 25 | 25 | 25 | 9 | 9 | 9 |
| Relative positional encoding padding (Transformer) | 6 | 6 | 6 | 12 | 12 | 12 | 4 | 4 | 4 |
| Relative positional encoding stride (Transformer) | 1 | 1 | 1 | 1 | 1 | 1 | 1 | 1 | 1 |
| Masking start prob ($p_m$) | 0.15 | 0.15 | 0.15 | 0.065 | 0.065 | 0.065 | 0.1 | 0.1 | 0.1 |
| Mask length ($m_l$) | 3 | 3 | 3 | 10 | 10 | 10 | 3 | 3 | 3 |
| Teacher model, initial weight update rate ($\tau_0$) | N/A | 0.9998 | N/A | N/A | 0.9995 | N/A | N/A | 0.9995 | N/A |
| Teacher model, final weight update rate ($\tau_{end}$) | N/A | 0.99999 | N/A | N/A | 0.99999 | N/A | N/A | 0.99999 | N/A |
| Teacher model, num weight update rate transitions ($\tau_n$) | N/A | 10,000 | N/A | N/A | 10,000 | N/A | N/A | 20,000 | N/A |

Table 5: The fine-tuning hyperparameters for PFML, data2vec, and MAE pre-trained models for each data modality (IMU, speech, and EEG data).

| | Multi-sensor IMU data | Speech data | EEG data |
|---|---|---|---|
| Patience (epochs) | 100 | 50 | 50 |
| Initial LR | 4e-5 | 4e-5 | 4e-5 |
| LR scheduler patience (epochs) | 30 | 15 | 15 |
| LR scheduler reduction factor | 0.5 | 0.5 | 0.5 |
| Optimization algorithm | Adam | Adam | Adam |
| Minibatch size | 1 | 16 | 128 |
| Loss function | Weighted cross-entropy | Weighted cross-entropy | Weighted cross-entropy |
| Dropout (encoder) | 0.3 | 0.3 | 0.1 |
| Dropout (Transformer) | 0.4 | 0.3 | 0.2 |

## C   RESULTS ON ADDITIONAL HYPERPARAMETER EXPERIMENTS

This section provides the result tables for the additional hyperparameter experiments in Section 4.5. Table 6 presents the fine-tuning results for the comparison between masking either the model inputs or embeddings during PFML pre-training. Tables 7, 8, and 9 show the fine-tuning results for different configurations of masking probabilities ($p_m$) and mask lengths ($m_l$) for multi-sensor IMU, speech, and EEG data, respectively. Table 10 shows the fine-tuning results for discarding some of the 11 functionals in PFML pre-training for multi-sensor IMU data. Finally, Table 11 presents the fine-tuning results for different mask types for PFML pre-training for multi-sensor IMU data.

Table 6: The fine-tuning results for the comparison between masking either inputs or embeddings for PFML pre-training.

| | Multi-sensor IMU data | | Speech data | | EEG data |
|---|---|---|---|---|---|
| | Movement | Posture | Valence | Arousal | Sleep stage |
| *Inputs masked* | 81.4 | 95.6 | 69.8 | 67.9 | **71.2** |
| *Embeddings masked* | **81.8** | **95.7** | **70.7** | **68.6** | 71.2 |
| | UAF1 (%) | | UAR (%) | | UAF1 (%) |

Table 7: The fine-tuning results for different PFML pre-training masking hyperparameter configurations for multi-sensor IMU data. The results are shown both with a fixed mask length and varying masking probability (left), and vice versa (right).

| Masking start prob ($p_m$) | Mask length ($m_l$) | Movement (UAF1 %) | Masking start prob ($p_m$) | Mask length ($m_l$) | Movement (UAF1 %) |
|---|---|---|---|---|---|
| 0.11 | 6 | 81.3 | 0.08 | 7 | 81.0 |
| 0.08 | 6 | 81.3 | 0.08 | 6 | 81.3 |
| 0.05 | 6 | 81.3 | 0.08 | 5 | 81.1 |
| 0.16 | 5 | 81.3 | 0.13 | 6 | 81.1 |
| 0.13 | 5 | 81.5 | 0.13 | 5 | 81.5 |
| 0.10 | 5 | 81.1 | 0.13 | 4 | 81.5 |
| 0.15 | 4 | 81.6 | 0.12 | 5 | 81.4 |
| 0.12 | 4 | 81.6 | 0.12 | 4 | 81.6 |
| 0.09 | 4 | 81.4 | 0.12 | 3 | 81.6 |
| 0.20 | 3 | 81.4 | 0.15 | 4 | 81.6 |
| 0.15 | 3 | **81.8** | 0.15 | 3 | **81.8** |
| 0.10 | 3 | 81.5 | 0.15 | 2 | 81.2 |
| 0.29 | 2 | 81.5 | 0.22 | 3 | 81.3 |
| 0.22 | 2 | 81.6 | 0.22 | 2 | 81.6 |
| 0.15 | 2 | 81.2 | 0.22 | 1 | 81.2 |
| 0.59 | 1 | 81.3 | 0.49 | 2 | 80.9 |
| 0.49 | 1 | 81.6 | 0.49 | 1 | 81.6 |
| 0.39 | 1 | 81.4 | | | |

Table 8: The fine-tuning results for different PFML pre-training masking hyperparameter configurations for speech data. The results are shown both with a fixed mask length and varying masking probability (left), and vice versa (right).

| Masking start prob ($p_m$) | Mask length ($m_l$) | Valence (UAR %) | Masking start prob ($p_m$) | Mask length ($m_l$) | Valence (UAR %) |
|---|---|---|---|---|---|
| 0.053 | 14 | 69.2 | 0.048 | 16 | 69.0 |
| 0.048 | 14 | 69.7 | 0.048 | 14 | 69.7 |
| 0.043 | 14 | 69.7 | 0.048 | 12 | 70.4 |
| 0.061 | 12 | 70.1 | 0.055 | 14 | 69.1 |
| 0.055 | 12 | 70.6 | 0.055 | 12 | 70.6 |
| 0.049 | 12 | 70.5 | 0.055 | 10 | 70.3 |
| 0.072 | 10 | 70.5 | 0.065 | 12 | 70.1 |
| 0.065 | 10 | **70.7** | 0.065 | 10 | **70.7** |
| 0.058 | 10 | 70.4 | 0.065 | 8 | 69.4 |
| 0.08 | 8 | 70.4 | 0.07 | 10 | 70.5 |
| 0.07 | 8 | 69.4 | 0.07 | 8 | 69.4 |
| 0.06 | 8 | 69.3 | 0.07 | 6 | 70.0 |
| 0.11 | 6 | 69.8 | 0.09 | 8 | 70.5 |
| 0.09 | 6 | 70.1 | 0.09 | 6 | 70.1 |
| 0.07 | 6 | 70.0 | 0.09 | 4 | 69.5 |
| 0.25 | 4 | 69.8 | 0.20 | 6 | 69.5 |
| 0.20 | 4 | 70.2 | 0.20 | 4 | 70.2 |
| 0.15 | 4 | 69.8 | 0.20 | 2 | 69.6 |

Table 9: The fine-tuning results for different PFML pre-training masking hyperparameter configurations for EEG data. The results are shown both with a fixed mask length and varying masking probability (left), and vice versa (right).

| Masking start prob ($p_m$) | Mask length ($m_l$) | Sleep stage (UAF1 %) | Masking start prob ($p_m$) | Mask length ($m_l$) | Sleep stage (UAF1 %) |
|---|---|---|---|---|---|
| 0.10 | 4 | 70.3 | 0.07 | 5 | 70.0 |
| 0.07 | 4 | 70.6 | 0.07 | 4 | 70.6 |
| 0.04 | 4 | 70.1 | 0.07 | 3 | 71.0 |
| 0.15 | 3 | 70.8 | 0.10 | 4 | 70.3 |
| 0.10 | 3 | **71.2** | 0.10 | 3 | **71.2** |
| 0.05 | 3 | 71.0 | 0.10 | 2 | 70.6 |
| 0.32 | 2 | 70.9 | 0.25 | 3 | 70.5 |
| 0.25 | 2 | 71.0 | 0.25 | 2 | 71.0 |
| 0.18 | 2 | 70.8 | 0.25 | 1 | 70.3 |
| 0.49 | 1 | 70.4 | 0.40 | 2 | 70.8 |
| 0.40 | 1 | 70.6 | 0.40 | 1 | 70.6 |
| 0.31 | 1 | 70.4 | | | |

Table 10: The fine-tuning results for discarding some of the functionals in PFML pre-training for IMU data.

| Num functionals | Functionals left out | Movement (UAF1 %) |
|---|---|---|
| 11 | − | **81.8** |
| 9 | min, max | 81.7 |
| 7 | min, max, ACF skewness, ACF kurtosis | 81.4 |
| 5 | min, max, ACF variance, ACF skewness, ACF kurtosis, ZCR | 81.0 |

Table 11: The fine-tuning results for different mask types for PFML pre-training for IMU data.

| Mask type | Movement (UAF1 %) |
|---|---|
| zeros | 81.2 |
| ones | **81.8** |
| Gaussian noise | 81.7 |
| learnable mask | 81.5 |

# D ADDITIONAL INFORMATION ON COMPUTATIONAL RESOURCES

All computations were run on a computing cluster operating on the SLURM environment. For both model pre-training and fine-tuning, we used an NVIDIA Tesla V100 GPU with 16 GB of VRAM, four CPU cores, and 16 GB of RAM. Table 12 shows the pre-training durations for each data modality (multi-sensor IMU data, speech data, and EEG data). Note that due to RAM constraints, samples of each minibatch were separately loaded from a disk to RAM. The possibility to load the entire training dataset into RAM would speed up the pre-training process substantially.

Table 12: The PFML pre-training durations for each data modality (IMU, speech, and EEG data).

| Data modality | Pre-training time (hours) |
|---|---|
| Multi-sensor IMU data | 21.6 |
| Speech data | 32.7 |
| EEG data | 16.1 |

The research project required more computations than what is reported in the present paper: For each data modality and pre-training algorithm, preliminary experiments were conducted in order to find suitable hyperparameters for both the pre-training and fine-tuning processes (Tables 4 and 5, respectively).

