# OpenReview forum: "PFML: Self-Supervised Learning of Time-Series Data Without Representation Collapse"
_ICLR.cc/2025/Conference — ICLR 2025 Conference Withdrawn Submission_

### Official Review · Reviewer_66Kv · 2024-11-02

**Soundness:** 2
**Presentation:** 2
**Contribution:** 2
**Rating:** 3
**Confidence:** 3

**Summary:**

The "PFML" paper presents a self-supervised learning (SSL) framework designed to address common challenges in time-series data, particularly avoiding representation collapse. The proposed model, Prediction of Functionals from Masked Latents (PFML), predicts statistical functionals rather than directly reconstructing the masked input signals. This approach is tested across three time-series data modalities—infant movement classification, emotion recognition, and sleep stage classification—demonstrating competitive performance and enhanced robustness against representation collapse compared to other SSL methods.

**Strengths:**

1.	PFML introduces an approach to SSL by predicting statistical functionals, effectively avoiding representation collapse, which is a common limitation in existing methods.
2.	The method requires minimal hyperparameter tuning, making it more accessible for new data domains and modalities. Additionally, PFML shows strong resistance to representation collapse, as demonstrated in multiple data contexts.
3.	PFML performs well across different time-series modalities, indicating broad potential for application in various fields such as healthcare and speech analysis.

**Weaknesses:**

1.	The motivation for using functionals as a target, while somewhat innovative, should be detailed to clarify its advantages over other SSL strategies, such as contrastive learning or clustering, in various time-series data types.
2.	While PFML shows competitive results against baseline SSL methods, a more extensive comparison with recent SSL techniques for time-series data would better illustrate its relative strengths and areas for improvement.
3.	The theoretical foundation of PFML is relatively limited, with insufficient rigorous derivation to support why predicting statistical functionals prevents representation collapse more effectively than other SSL methods. A more formal analysis would strengthen the paper and clarify the principles driving PFML’s effectiveness.
I will reconsider my assessment after checking the rebuttal.

**Questions:**

1.	Could the authors provide more insight into how the choice of statistical functionals impacts PFML's adaptability across diverse time-series tasks?
2.	How does PFML handle extremely noisy or incomplete data segments during pre-training?
3.	Could this approach potentially extend to other types of sequential data, such as video or more structured clinical data like ECG?

---

### Official Review · Reviewer_nsVC · 2024-11-02

**Soundness:** 1
**Presentation:** 2
**Contribution:** 1
**Rating:** 3
**Confidence:** 5

**Summary:**

The paper proposes a new style of self-supervised learning (SSL) with the core goal of avoiding representation collapse. It introduces a somewhat novel proxy task, where the model predicts statistical functionals of the masked segments instead of simply predicting the masked segment values. This approach aims to strike a balance between task simplicity and complexity. Their main claim is that this conceptually simple method effectively addresses the issue of representation collapse for time-series data.

**Strengths:**

1. The paper’s motivation is highly practical and deserves more attention from the time-series machine learning community.
2. The simplicity-first design intent should be well-regarded in the machine learning community, and the paper supports this approach effectively.

**Weaknesses:**

1. The introduction of the paper misses several important references in the field of self-supervised learning including the very important non-contrastive methods since that is the whole premise of their motivation -- avoiding the choice of negative and positive anchors for contrastive learning [1]. In fact, previous works [2] have successfully leveraged SimSiam for audio/speech applications, which is one of the benchmark datasets in this work as well.

2. I am curious about how the authors construct the functionals for multivariate time-series inputs.

3. Lines 255-258 are challenging to follow, as the notations are not adequately explained. For example, in \( m_l \), what does \( l \) represent?

4. Several claims are made without experimental substantiation:
   - **4.1.** The authors state that the number of hyperparameters is fewer than in SOTA methods, but there is no objective data to support this. Also, the success of this method seems highly dependent on the window size for the masked embedding and the overall stochasticity of the underlying dataset.
   - **4.2.** Echoing some previous concerns, Assumption 1 (lines 236-237) is overly strong and cannot serve as the foundation of this method. It is unreasonable to expect the dataset to present sufficient variability/nonstationarity to extract meaningful functionals and prevent collapse. What if I want to model a direct current (DC)-like signal, which is a component of my overall dataset?
   - **4.3.** Some theoretical grounding to support the claim of avoiding collapse would increase confidence in this method’s applicability to general time-series data.

5. The authors only consider two baselines. To genuinely claim the superiority of their method, they should evaluate it against several prominent time-series baselines Ts2Vec [3], BYOL [4] etc. Additionally, since the claim pertains to SSL representation learning, incorporating more diverse tasks (forecasting, anomaly detection, periodic target detection etc.) and datasets would help readers assess the method's utility for their applications.


[1] "Exploring simple siamese representation learning." In Proceedings of the IEEE/CVF conference on computer vision and pattern recognition, 2021.

[2] "Efficient stuttering event detection using siamese networks." In ICASSP 2023-2023 IEEE International Conference on Acoustics, Speech and Signal Processing (ICASSP), 2023.

[3] "Ts2vec: Towards universal representation of time series." In Proceedings of the AAAI Conference on Artificial Intelligence, 2022.

[4] "Bootstrap your own latent-a new approach to self-supervised learning." Advances in neural information processing systems 33 (2020).

**Questions:**

See weaknesses above.

---

### Official Review · Reviewer_dXXB · 2024-11-03

**Soundness:** 1
**Presentation:** 2
**Contribution:** 1
**Rating:** 3
**Confidence:** 4

**Summary:**

The paper presents PFML (Prediction of Functionals from Masked Latents), a self-supervised learning (SSL) algorithm designed to handle time-series data without suffering from representation collapse. The approach focuses on predicting statistical functionals (e.g., mean, variance) from masked embeddings instead of reconstructing the input signal, demonstrating comparable performance over similar SSL methods like MAE and Data2Vec in classification tasks for infant movement, speech emotion, and EEG-based sleep stages.

**Strengths:**

- Novelty in SSL for Time-Series: PFML introduces an innovative approach by predicting statistical functionals, reducing the complexity of reconstructing time-series data, a significant improvement over existing SSL methods.
- Effective Across Multiple Modalities: The method is shown to work well across various domains, including IMU, speech, and EEG data, demonstrating its flexibility and robustness.
- Simplicity in Hyperparameter Tuning: Unlike many SSL algorithms that require complex tuning, PFML is designed with simplicity, reducing the trial-and-error process in hyperparameter optimization.
- Avoids Representation Collapse: PFML addresses a critical challenge in SSL, representation collapse, proving its reliability in training without requiring extensive countermeasures.

**Weaknesses:**

- Limited Exploration of Functionals: While PFML utilizes a set of many functionals, the choice of these functionals is somewhat arbitrary, and there is limited discussion on how different sets of functionals might affect performance across various datasets. As a matter of fact, many of the functionals are permutation-invariant like mean, variance, skewness and kurtosis directly computed from input data, while those computed from autocorrelation function can also be easily matched with a carefully permuted input data. This raises concerns about the technical soundness of the method, and it sets may not preserve the temporal structure of the data, which is crucial for time-series data.
- No Data Augmentation: Despite recognizing the potential benefits of data augmentation during pre-training, the authors did not implement it to save computational time, which might limit the generalizability of results.
- Narrow Benchmark Comparisons: The paper primarily compares PFML with MAE and Data2Vec. It would benefit from a broader range of comparisons with more diverse SSL methods across different tasks.
- Lack of Theoretical Depth in Certain Areas: The theoretical justification for using functionals is not fully explored in terms of how it scales with different kinds of temporal variations or non-stationary data, potentially limiting understanding of the full potential of PFML.

**Questions:**

1. Could you provide an ablation study on the choice of functionals to demonstrate the impact of different sets of functionals on the performance of PFML across various datasets? Through comparison, it seems that the improvement in performance is marginal, which raises concerns about the necessity of using a large set of functionals and the potential concerns regarding effectiveness.
2. Could you also provide additional comparison studies into the effectiveness of the proposed method for stationary and non-stationary data, and preferably include some synthetic datasets as well to study the design choice of functionals for different characteristics of data such as seasonality, trend, auto correlation, etc.?
3. You proposed that representation collapse is critical in SSL, yet the paper does not provide a detailed discussion on how PFML effectively mitigates this issue, as MAE seems to not suffer from representation collapse already. And although data2vec do suffer from this issue, it is not clear how PFML is better than data2vec in this regard. Could you provide more insights into this? How do MAE and data2vec behave differently when one is experiencing representation collapse and the other isn't? Perhaps some visualization will help.
4. The current evaluation of the method seems to be limited, in both the number of datasets and baselines. Could you provide more comprehensive evaluations on a broader range of datasets and SSL methods to better understand the generalizability of PFML? Such as, how does PFML compare to other SSL methods like [TimeCLR](https://doi.org/10.1016/j.knosys.2022.108606), [Time2Vec](https://arxiv.org/abs/1907.05321) and [TF-C](https://proceedings.neurips.cc/paper_files/paper/2022/hash/194b8dac525581c346e30a2cebe9a369-Abstract-Conference.html)?

---

### Official Review · Reviewer_4Sky · 2024-11-05

**Soundness:** 3
**Presentation:** 3
**Contribution:** 2
**Rating:** 3
**Confidence:** 4

**Summary:**

This paper introduces a novel framework called Prediction of Functionals from Masked Latents (PFML) for modeling time-series
data in self-supervised learning (SSL) pre-training. PFML stands out by eliminating the need for careful hyperparameter selection
and mitigating representation collapse. Experimental results demonstrate that PFML outperforms a comparable SSL method and
achieves competitive performance against SOTA SSL approaches.

**Strengths:**

1. The PFML method proposed in this paper effectively eliminates the cumbersome process of selecting hyperparameters, allowing the same model to be applied across different domains without the need for adjustments.
2. The PFML model successfully avoids representation collapse during SSL pre-training, preventing the model from outputting a constant, time invariant representation. The paper provides detailed proof to support
this claim.
3. In the experimental section, the paper offers a thorough analysis of hyperparameters, the number of statistical functions, and the computational resources required for the proposed method.

**Weaknesses:**

1. The model proposed in this paper is an adaptation of the MAE model. The masking mechanism draws inspiration from wav2vec and data2vec, aiming to reconstruct the statistical information of the input signal. However, these contributions seem limited in scope.
2. The experimental results of this model are only comparable to data2vec, which was published two years ago, suggesting that its performance may not surpass more recent methods in the field.

**Questions:**

1.Why were experiments conducted only on CNN and Transformer models, when the paper mentions that PFML is
not limited to any specific model and can be applied to multiple sequence encoding models?
2.The experimental section could be expanded to include additional results showcasing the application of PFML to other models, thereby demonstrating the broader applicability and universality of the PFML framework.

---

### Note · Authors · 2024-11-18

I have read and agree with the venue's withdrawal policy on behalf of myself and my co-authors.